# mRNA booster vaccination protects aged mice against the SARS-CoV-2 Omicron variant

Etsuro Nanishi [1,2,11], Marisa E. McGrath [3,11], Timothy R. O'Meara [1], Soumik Barman [1,2], Jingyou Yu[4], Huahua Wan[4], Carly A. Dillen[3], Manisha Menon[1], Hyuk-Soo Seo [5,6], Kijun Song [5], Andrew Z. Xu [5], Luke Sebastian [5], Byron Brook [1,2], Anna-Nicole Bosco[1], Francesco Borriello[1,2,7,10], Robert K. Ernst [8], Dan H. Barouch [4], Sirano Dhe-Paganon [5,6], Ofer Levy [1,2,9,12], Matthew B. Frieman [3,12] & David J. Dowling [1,2,12]✉

The SARS-CoV-2 Omicron variant evades vaccine-induced immunity. While a booster dose of ancestral mRNA vaccines effectively elicits neutralizing antibodies against variants, its efficacy against Omicron in older adults, who are at the greatest risk of severe disease, is not fully elucidated. Here, we evaluate multiple longitudinal immunization regimens of mRNA BNT162b2 to assess the effects of a booster dose provided >8 months after the primary immunization series across the murine lifespan, including in aged 21-month-old mice. Boosting dramatically enhances humoral and cell-mediated responses with evidence of Omicron cross-recognition. Furthermore, while younger mice are protected without a booster dose, boosting provides sterilizing immunity against Omicron-induced lung infection in aged 21-month-old mice. Correlational analyses reveal that neutralizing activity against Omicron is strongly associated with protection. Overall, our findings indicate age-dependent vaccine efficacy and demonstrate the potential benefit of mRNA booster immunization to protect vulnerable older populations against SARS-CoV-2 variants.

[1] Precision Vaccines Program, Division of Infectious Diseases, Boston Children's Hospital, Boston, MA, USA. [2] Department of Pediatrics, Harvard Medical School, Boston, MA, USA. [3] Department of Microbiology and Immunology, The Center for Pathogen Research, University of Maryland School of Medicine, Baltimore, MD, USA. [4] Center for Virology and Vaccine Research, Beth Israel Deaconess Medical Center, Harvard Medical School, Boston, MA, USA. [5] Department of Cancer Biology, Dana-Farber Cancer Institute, Boston, MA, USA. [6] Department of Biological Chemistry and Molecular Pharmacology, Harvard Medical School, Boston, MA, USA. [7] Division of Immunology, Boston Children's Hospital, Boston, MA, USA. [8] Department of Microbial Pathogenesis, University of Maryland School of Dentistry, Baltimore, MD, USA. [9] Broad Institute of MIT & Harvard, Cambridge, MA, USA. [10] Present address: Generate Biomedicines, Cambridge, MA, USA. [11] These authors contributed equally: Etsuro Nanishi, Marisa E. McGrath. [12] These authors jointly supervised this work: Ofer Levy, Matthew B. Frieman, David J. Dowling. ✉email: david.dowling@childrens.harvard.edu

The SARS-CoV-2 Omicron variant was identified with multiple mutations in the spike protein and demonstrated escape from vaccine-induced neutralizing antibodies (Abs)[1–4]. The Omicron variant rapidly became predominant and induced a sharp rise of infections in populations with a high prevalence of SARS-CoV-2 immunity[5]. Administering a booster dose of mRNA vaccines coding for the ancestral wildtype (WA-1) spike protein induces neutralizing Abs against variants and thus represents an approach to combat the emergence of new variants. However, whether a booster can provide protective efficacy against Omicron in older individuals, who have the greatest risk of severe infection and mount less effective immune responses due to immunosenescence[6], had not been investigated. Here, we evaluated multiple immunization regimens of mRNA BNT162b2 to evaluate the effect of a booster dose in aged 21-month-old mice. Overall, we demonstrated age-dependent vaccine efficacy, with mRNA booster immunization being essential to protect aged mice against lower respiratory infection from SARS-CoV-2 variants via dramatically enhanced immune responses that cross-recognize Omicron.

## Results

To evaluate age-specific differences in short- and long-term immunogenicity of the ancestral mRNA vaccines, we first immunized 3-month-old and 11-month-old BALB/c mice ($N = 40$ per group) with 1 µg of mRNA BNT162b2 (Supplementary Fig. 1). The primary vaccination series elicited lower anti-spike Ab titers in 11-month-old mice compared to 3-month-old mice. Notably, humoral immune responses waned drastically in older mice as shown by a marked decline in hACE2-RBD inhibition (median of 95.0% and 14.4% at week 4 and 32, respectively), while younger mice maintained a high inhibition rate (99.6% and 97.2% at week 4 and 32, respectively) (Supplementary Fig. 1d, e). Next, to address whether a booster dose can restore waning immunity in older mice, we randomly assigned mice to the following groups ($N = 14–20$ per group): mock PBS injection (naive); a primary mRNA vaccination series at weeks 32–34 (recent vax); a primary vaccination series at weeks 0–2 without a booster dose (distant vax); and a primary vaccination series at weeks 0–2 with a booster dose at week 34 (booster vax) (Fig. 1). Additionally, 3-month-old BALB/c mice were added to the study at week 32 and allocated to naive or recent vax groups. Serum samples were collected at week 36 and Ab binding to the vaccinal antigen (prefusion form of wildtype spike trimer) was assessed by ELISA. As expected, increased age was associated with a significant decline in binding Ab titers among both mice that received the primary vaccination series <1 month (recent vax) and the series >8 months ago (distant vax). Comparisons between 5-month-old recent versus 13-month-old distant, and 13-month-old recent versus 21-month-old distant mice allowed us to analyze waning immunity in relation to aging since the mice were immunized at same ages (3 and 11 months old, respectively) and followed for different periods (2 and 10 months, respectively) (Fig. 1). Interestingly, although 5-month-old recent vax and 13-month-old distant vax showed comparable spike-specific IgG titers, 21-month-old distant vax showed significantly lower titers compared to 13-month-old recent vax (Fig. 2a). This result further indicates that increasing age is associated with greater waning of humoral responses. A booster dose dramatically enhanced humoral responses and elicited significantly higher Ab titers versus recent and distant groups among 13- and 21-months age (Fig. 2a). Notably, 21-month-old mice demonstrated non-inferior spike-specific IgG responses compared to younger mice after receiving a booster dose. These results were consistent in a hACE2-RBD binding inhibition assay (Fig. 2b, c).

### a. Mouse age groups

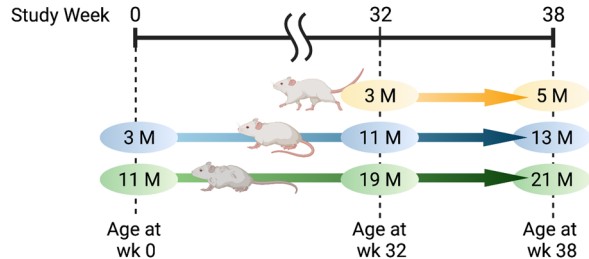

### b. Treatments within each age group

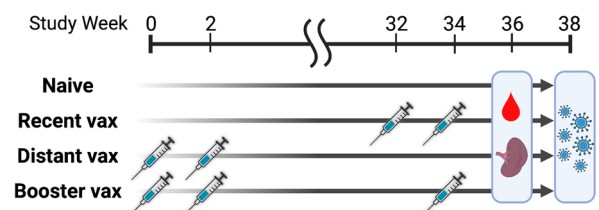

**Fig. 1 Study design outlining longitudinal immunization of mRNA BNT162b2 across the lifespan.** Schematic representation of the study design showing three mouse age groups (**a**) and treatments within each age group (**b**). Three- and 11-month-old BALB/c mice were included in the study at week 0 and received the following treatments: mock PBS injection (naive); primary mRNA vaccination series at weeks 32–34 (recent vax); primary vaccination series at weeks 0–2 (distant vax); and primary vaccination series at weeks 0–2 with booster dose at week 34 (booster vax). Additionally, 3-month-old BALB/c mice were enrolled in the study at week 32 and received the following treatments: mock PBS injection (naive); and primary vaccination series at weeks 32–34 (recent vax). $N = 20$ per group at enrollment. Serum samples were collected at week 36. Splenocytes were collected at week 36 from $N = 4–7$ mice per group. Mice were challenged with SARS-CoV-2 Omicron strain at week 38 ($N = 5–9$ per group). The graphics were created with BioRender.com.

Neutralizing Abs (NAbs) are important for protecting from SARS-CoV-2 infection[7–9]. We therefore evaluated serum neutralizing responses by pseudovirus assays. All samples demonstrated neutralizing activity against the vaccine-matched wildtype strain. As with spike-specific IgG titers, NAb titers decreased with increasing age. A booster dose dramatically enhanced NAb titers, with aged mice demonstrating non-inferior responses versus younger mice (Fig. 2d). A marked reduction of serum neutralizing activity against SARS-CoV-2 variants was observed in pooled samples, with geometric mean titers (GMT) of 2281 (wildtype), 748 (Delta), and 83 (Omicron) (Fig. 2e). Neutralizing titers against Omicron remained low in all age groups after two-dose vaccination. Notably, none of the 21-month-old distant vax mice demonstrated detectable neutralizing titers (Fig. 2d, e). Strikingly, however, neutralizing activity against Omicron dramatically increased after a booster dose and was detectable in all 21-month-old mice. These data indicate that a booster dose is very effective at enhancing humoral immune responses and eliciting NAbs against the Omicron variant even in aged mice.

T cell responses induced by SARS-CoV-2 vaccines suppress viral replication and modulate disease severity[9–11]; however, it has remained unclear whether T cell responses decrease months after a primary immunization series, especially in older populations, and if so whether a booster dose can restore cellular immunity. To address these questions, we analyzed specific T cell responses at week 36 as shown in Fig. 1. Splenocytes were collected from immunized mice [$N = 6–7$ per group, except for 21 M

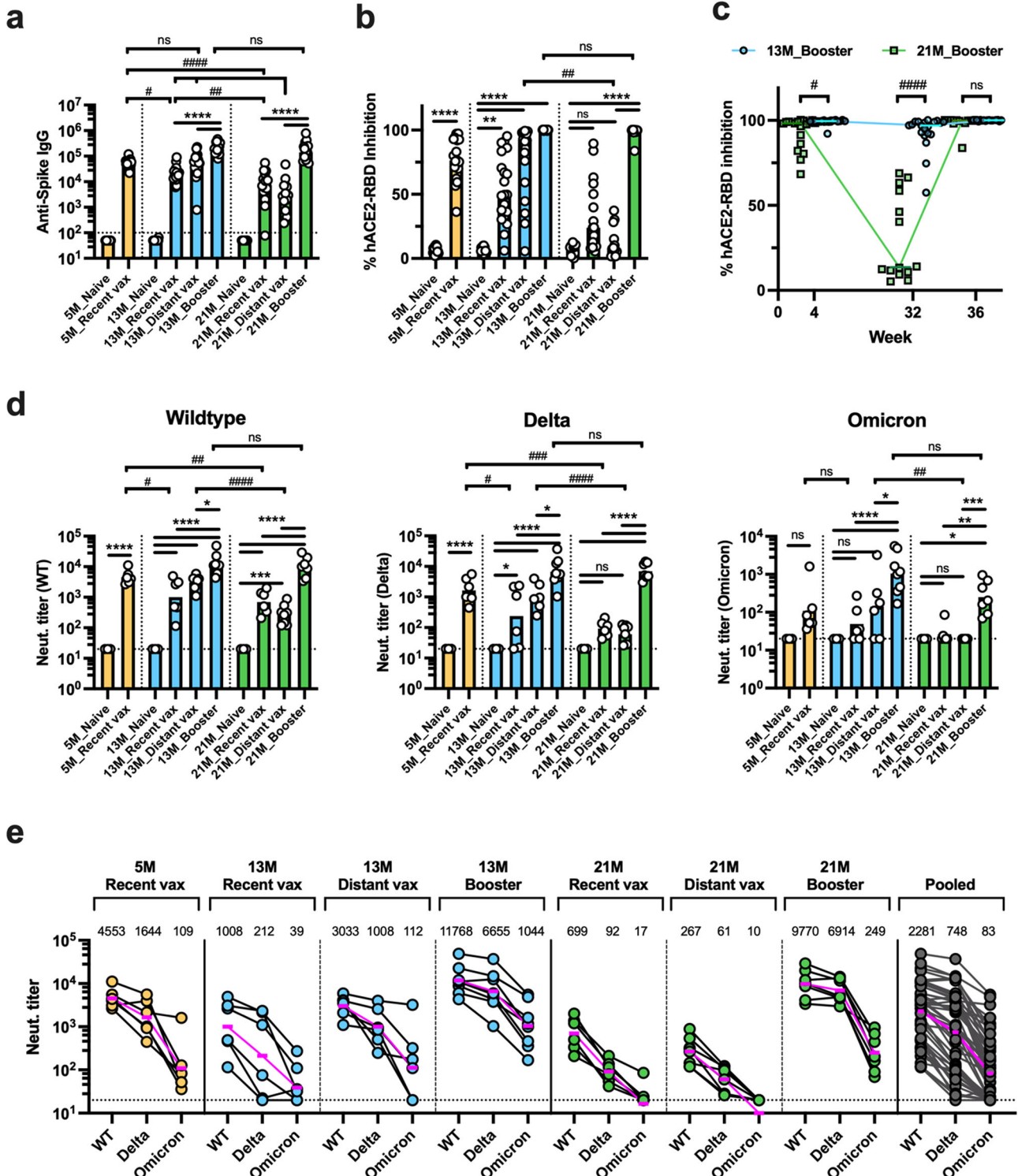

**Fig. 2 A booster dose of mRNA BNT162b2 restores waning humoral responses in aged mice and elicits potent neutralizing activity against SARS-CoV-2 Omicron.** Sera were collected from immunized mice at week 36 to determine humoral responses as indicated in Fig. 1. Anti-WT spike IgG titers (**a**), hACE2-RBD (WT) inhibition rate (**b**, **c**), and neutralization titers against WT, Delta, and Omicron pseudoviruses (**d**, **e**) were determined. Results are presented as individual values with mean (**a**, **d**) or median (**b**, **c**). Dashed lines represent lower limit of detection. Data were analyzed by one-way ANOVA after log-transformation (**a**, **d**) or Kruskal–Wallis test (**b**, **c**) corrected for multiple comparisons. * indicates comparisons within the same age group and # indicates comparisons between different age groups. (**a**–**c**) $N = 19$–20 per group for 5 and 13 M aged mice, and $N = 13$–17 per group for 21 M aged mice. **e** Horizontal lines and numbers indicate geometric mean titers (GMTs). For GMT calculation, an arbitrary value of 10 was assigned to samples with values below the detection limit of 20. Dashed lines represent lower limit of detection. **d**, **e** $N = 6$–8 per group except for Naive mice ($N = 3$).*/#$P < 0.05$, **/##$P < 0.01$, ***/###$P < 0.001$, ****/####$P < 0.0001$.

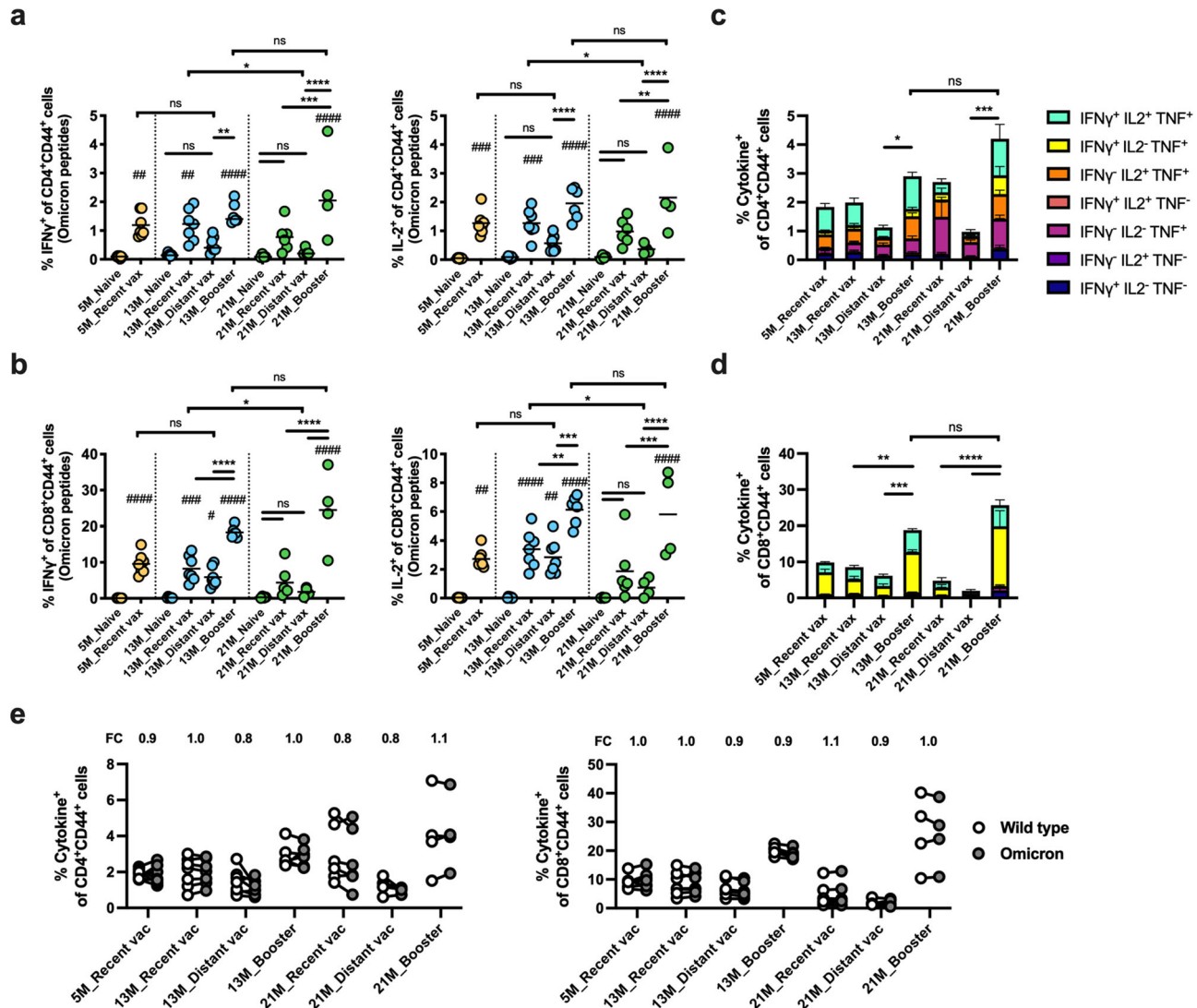

**Fig. 3 A booster dose of mRNA BNT162b2 elicits robust T cell responses that cross-recognize SARS-CoV-2 Omicron in aged mice.** SARS-CoV-2 spike specific T cell responses were analyzed at week 36 as indicated in Fig. 1. Splenocytes were restimulated with overlapping spike peptides from Omicron (**a**–**e**) and wildtype (**e**) SARS-CoV-2, and expression of intracellular interferon-γ (IFNγ), IL-2, and TNF among CD4+ and CD8+ T cells were determined by flow cytometry. $N = 6$–7 per group, except for 21 M Distant vax and Booster groups ($N = 4$). **a**, **b** Frequencies of IFNγ and IL-2 positive T cells among CD4+ (**a**) and CD8+ (**b**) T cells are shown. **c**, **d** Frequency of mono-, double-, and triple- cytokine positive cells among CD4+ (**c**) and CD8+ (**d**) T cells are summarized. Lines represent mean ± SEM. Data were analyzed by one-way ANOVA corrected for multiple comparisons. # indicates for comparisons against naïve groups in the same age group. */#$P < 0.05$, **/##$P < 0.01$, ***/###$P < 0.001$, ****/####$P < 0.0001$. **e** Frequency of cytokine positive (expressing any of IFNγ, IL-2, and/or TNF) CD4+ and CD8+ T cells against wildtype or Omicron spike peptide pools were assessed. Each symbol represents one mouse. Omicron/wildtype fold change (FC) values were calculated, and the geometric mean of FC values are listed at the top of each graph.

distant vax and booster groups ($N = 4$)] and restimulated with overlapping peptides spanning the full length of the Omicron or wildtype spike protein. Intracellular expression of interferon-γ (IFNγ), IL-2, and TNF among CD4+ and CD8+ T cells were assessed to quantify spike-specific T cell responses (Supplementary Fig. 2). Following stimulation with Omicron spike peptides, significant decreases in IFNγ and IL-2 expressing T cells were observed in 21-month-old distant vax compared to 13-month-old recent vax mice but not in the younger groups (5-month-old recent vax versus 13-month-old distant vax) (Fig. 3a, b). These results indicate that increasing age is associated with greater waning of cell-mediated immune responses. Indeed, 21-month-old distant vax mice that had received the primary vaccination series >8 months prior showed the lowest spike-specific T cell frequency. Of note, a booster dose dramatically increased Omicron spike-specific CD4+ and CD8+ T cell responses compared

to age-matched mice that did not receive a booster dose (fold increase of mean frequencies: 2.6 and 4.3 for CD4+ T cells and 3.1 and 12.9 for CD8+ T cells among 13- and 21-month-old mice, respectively) (Fig. 3a–d). To assess the cross-reactivity of T cell responses, the magnitude of the SARS-CoV-2 spike-specific T cell responses induced by wildtype and Omicron peptide pools were compared. Remarkably, the T cell responses were mostly comparable across variants, regardless of mouse age and vaccine regimens. Geometric mean fold changes of Omicron/wildtype ranged from 0.79–1.07 for CD4+ T cells and 0.87–1.07 for CD8+ T cells (Fig. 3e). Overall, these data indicate that T cell responses induced by an ancestral wildtype spike-specific mRNA vaccine cross-recognize Omicron and that a booster dose dramatically enhances T cell responses in aged, 21-month-old mice.

To determine whether enhanced immune responses elicited by a booster dose translate to improved protection against severe

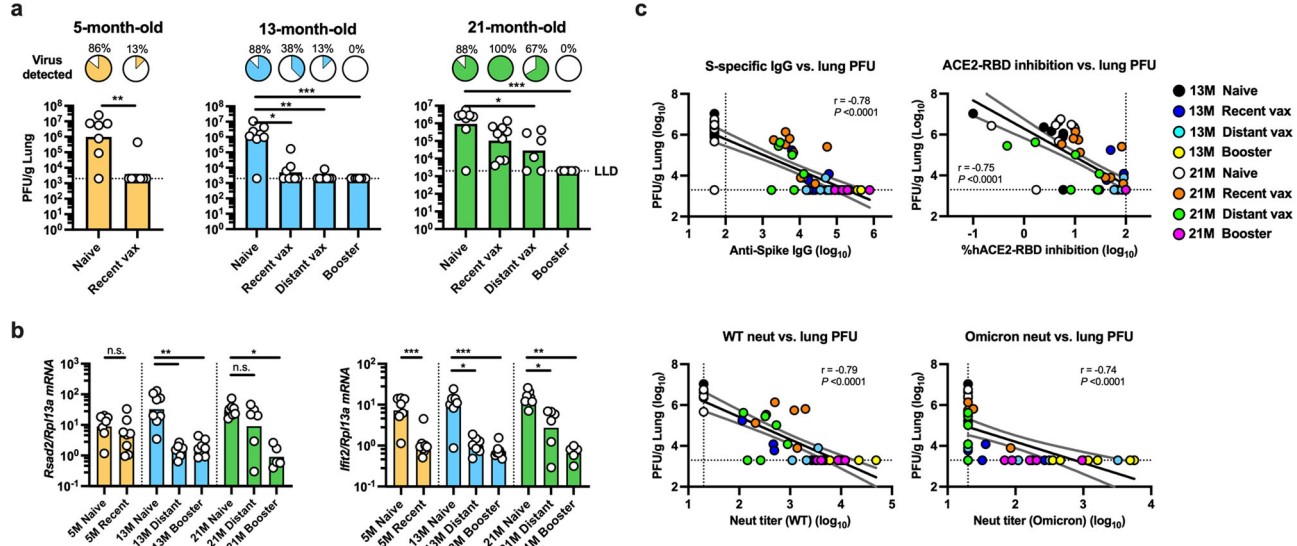

**Fig. 4 mRNA booster vaccination protects against lower respiratory infection by SARS-CoV-2 Omicron in aged mice.** SARS-CoV-2 mRNA BNT162b2-immunized 5-, 13-, and 21-month-old mice were challenged intranasally with $10^5$ plaque-forming units (PFU) of SARS-CoV-2 Omicron at week 38 as indicated in Fig. 1. Viral titers (**a**) and gene expression profiles of *Rsad2* and *Ifit2* shown as relative expression compared to *Rlp13a* (**b**) in lung homogenates at 2 days post challenge are shown. Bars represent geometric means. Pie charts show the proportion of mice within each group that had detectable virus in the lungs. Data were analyzed by Mann-Whitney test or Kruskal–Wallis test corrected for multiple comparisons. *$P < 0.05$, **$P < 0.01$, ***$P < 0.001$. $N = 7-9$ per group, except for 21 M Distant vax ($N = 6$) and 21 M Booster ($N = 5$) groups. **c** Correlations between lung PFU at 2 days post challenge and anti-spike IgG titer, hACE2-RBD inhibition rate, and pseudovirus neutralizing titers against SARS-CoV-2 WT and Omicron at week 36 are shown. Circles represent individual mice, and colors indicate the immunization regimens. Dotted lines indicate assay limits of detection. Black and gray lines respectively indicate linear regression and 95% confidence interval. Correlations were assessed by two-sided Spearman rank-correlation tests.

disease induced by Omicron (i.e., lower respiratory tract infection), we next assessed the protective efficacy in a murine challenge model. To this end, 5-, 13-, and 21-month-old mice that received multiple immunization regimens of mRNA BNT162b2 were intranasally challenged with $10^5$ PFU of SARS-CoV-2 Omicron at week 38, as indicated in Fig. 1 [$N = 7-9$ per group, except for 21 M distant vax ($N = 6$) and 21 M booster ($N = 5$) groups]. Viral titers in lung homogenates were assessed 2 days post-challenge. Naïve mice demonstrated robust lung viral loads, which were comparable across age groups (GMT: $9.97 \times 10^6$, $8.47 \times 10^6$, and $9.39 \times 10^6$ for 5-, 13-, and 21-month-old mice, respectively) (Fig. 4a). As expected, increased age was associated with lower vaccine efficacy. Among recent vax mice that received their primary vaccination series within 1 month of challenge, 6/8 and 4/8 of the 5- and 13-month-old mice, respectively, sterilized virus from the lungs, while all (9/9) of the 21-month-old mice demonstrated detectable viral titers (Fig. 4a). Importantly, none of the mice that received a booster dose demonstrated detectable live virus in the lung, regardless of age. In contrast, among distant vax mice that received the primary vaccination series >8 months before challenge, more than half (4/6) of the 21-month-old mice showed detectable viral loads, with a GMT of $2.89 \times 10^4$, while most (7/8) of 13-month-old mice demonstrated no live virus, with a GMT of $2.38 \times 10^3$ ($P = 0.04$) (Fig. 4a). These results indicate that a booster mRNA vaccine protects against severe disease caused by SARS-CoV-2 Omicron in aged, 21-month-old mice but less necessary in younger mice. Type I interferons are drivers of pathological responses after SARS-CoV-2 infection[12]. We therefore assessed the expression of IFN-stimulated genes (ISGs) and confirmed a decrease in the host antiviral response in the 21-month-old booster mice compared to the 21-month-old distant vax mice which would correlate to the lower viral load in the lungs (Fig. 4a, b). Lastly, to define immune correlates of protection against Omicron, we assessed correlations between lung viral loads at two days post-challenge and humoral immune readouts assessed at week 36. Anti-spike (WT) IgG, hACE2-RBD (WT) inhibition, and neutralizing titers against SARS-CoV-2 WT demonstrated strong inverse correlations with lung viral loads ($r = -0.78$, $-0.75$, and $-0.79$, respectively; all $P < 0.0001$) (Fig. 4b). Strikingly, NAbs against Omicron appeared to contribute to protection. All animals with neutralization titers of >100 against Omicron completely sterilized virus from the lungs, and even animals with low but detectable NAb levels still demonstrated low or undetectable lung viral loads (Fig. 4b).

## Discussion
In this study, we have demonstrated the age-specific impact of mRNA booster immunization on immunogenicity and protection in a murine model of SARS-CoV-2 infection. The severity of COVID-19 increases with age, while vaccine immunogenicity declines in older populations[6,13,14]. Although epidemiological data suggest that the Omicron variant induces milder disease compared to previous strains, morbidity and mortality in older adults remain high and represent a major health threat to this group[5,15,16]. SARS-CoV-2 mRNA vaccines initially showed remarkable efficacy and real-world effectiveness across age groups[17,18]. However, the emergence of immune-evasive variants along with declining immune memory following vaccination are associated with waning vaccine effectiveness (VE) and a corresponding resurgence in cases[19,20]. Although a booster dose of mRNA vaccines effectively overcame this waning VE in pre-Omicron variants[21], whether a booster dose of ancestral wildtype spike mRNA vaccines can improve VE against Omicron and whether the protective efficacy varies between age groups are unknown. Here, we assessed various immunization regimens in an age-specific mouse model to determine the effect of a booster dose, with an emphasis on older age. We demonstrated that a booster dose dramatically enhanced both humoral and T cell responses across age groups. Importantly, a booster dose provided sterilizing immunity against Omicron in aged mice, while mice

without a booster had insufficient immunity to clear virus form the lungs. In contrast, younger mice were protected from lung infection without a booster. Correlation analysis further revealed that NAbs against Omicron were strongly associated with protection. Additionally, boosting strategies demonstrated enhanced immunogenicity in aging and aged mice in other SARS-CoV-2 vaccine platforms, including adenoviral vector and adjuvanted protein subunit vaccines[22,23]. Overall, these results demonstrate the efficacy of a booster dose in aged populations and highlight the importance of a precision medicine approach to achieve protection in vulnerable populations with distinct immune responses.

Antibodies serve as a first line of defense for protection against SARS-CoV-2[7–9]. In this study, we show that a booster dose dramatically enhances humoral immune responses in multiple populations including aged 21-month-old mice. Of note, while increasing age was associated with decreased humoral immune responses after the primary vaccine series, 21-month-old mice demonstrated equivalent spike specific IgG and NAbs against wildtype SARS-CoV-2 compared to 13-month-old mice after receiving a booster dose. The Omicron variant substantially escapes from NAb responses[1–4]. In line with these studies, we demonstrated that NAbs induced by an ancestral mRNA vaccine show less inhibitory activity against Omicron. Although none of the 21-month-old mice showed detectable NAbs against Omicron >8 months after the primary immunization series, all of the 21-month-old mice demonstrated robust neutralizing activity after receiving a booster dose. The importance of NAbs was further confirmed in a correlation analysis with a clear threshold to achieve complete protection. While NAbs have been shown to predict the VE against wildtype and previous variants[7–9,24], it was unclear whether this correlation held true for Omicron. Our study adds important insight to the field by demonstrating that this correlation is consistently demonstrated with Omicron.

Vaccine-induced cellular immunity substantially contributes to protection against severe SARS-CoV-2 infection[9–11]. We demonstrated that while 21-month-old mice showed low levels of T cell responses >8 months after the primary vaccination series, a booster dose strikingly increased Omicron-spike-specific CD4$^+$ and CD8$^+$ T cell responses (4.3- and 12.9-fold increase, respectively), and these responses were comparable to those induced in younger mice that received a booster dose. Further, in line with the recent studies[25,26], we demonstrate that T cell responses induced by an ancestral mRNA vaccine cross-recognize the Omicron variant regardless of age and vaccine regimen. Since we had to euthanize mice and collect splenocytes to assess T cell response, our correlation analysis was limited to readouts of humoral immunity. However, T cells may play an important role in protecting against Omicron. Indeed, a few mice demonstrated protection from lung infection without detectable neutralizing activity. These findings suggest that protection against Omicron may correlate with both humoral and cellular immune responses.

Our study features several strengths, including (a) accounting for age-specific immunity that can play major roles in vaccine immunogenicity and protective efficacy, (b) accounting for both humoral and cell-mediated immunity reported to align best with known correlates of protection[10,27], and (c) performing a live SARS-CoV-2 Omicron challenge evaluating protective efficacy of employed vaccines. Our study also has some limitations. First, this study only contains mouse data, establishing the need for future translational research in additional animal models and humans. Second, due to the unique and extremely longitudinal nature of the study design, some of the experimental groups had a relatively low number of mice, restricting some statistical comparisons. Of note, a considerable number of aged mice developed conditions such as frailty and gross tumors over the 10-month study period, and only $N = 48$ out of the initial $N = 80$ could be

assessed for T cell responses ($N = 20$) or challenge study ($N = 28$). Nonetheless, due to the ample sample size for Ab analysis ($N = 13–17$ per group) and large effect size of the booster dose observed in T cell and challenge studies, the results of this study demonstrate the importance of booster vaccination in the aged 21-month-old mice. Third, due to limited availability, 3- and 11-month-old mice were purchased from different vendors. During this longitudinal study, $N = 40$ of initially 11-month-old mice received a primary vaccination series at week 0, while $N = 20$ of initially 3-month-old mice received a primary vaccination series at week 32 at the age of 11-month-old. We therefore compared immunogenicity between mice purchased from two vendors which received primary vaccination series at 11-month-old age and confirmed that the immunogenicity of 11-month-old mice from the two vendors were largely equivalent and both considerably different compared to 3-month-old mice (Supplementary Fig. 3). Accordingly, any vendor-related differences are unlikely to account for any of the key interpretations in our manuscript. Fourth, we only analyzed viral loads within lungs since our main focus was whether a booster dose can provide protection against severe infection (i.e., lower respiratory infection) by Omicron. However, whether a booster dose can suppress symptomatic and/or asymptomatic infection and subsequent transmission is also a key question to address. Similar studies focusing on viral replication in the upper respiratory tract are needed. Finally, we analyzed immunogenicity and protective efficacy at a single and relatively short timepoint. A longer time course analysis is needed to assess the durability of the enhanced immune responses induced by a booster dose.

Overall, our study evaluated the effect of a booster dose of an ancestral wildtype spike mRNA vaccine against SARS-CoV-2 Omicron across multiple ages, including in aged mice. A significant decline in vaccine-induced immune responses was observed in aged, 21-month-old mice >8 months after the primary vaccine series. A booster dose markedly enhanced both Ab and T cell responses with evidence for cross recognition of Omicron. Furthermore, we demonstrated that a booster dose was essential for protecting aged mice from severe Omicron infection. These results not only indicate that mRNA booster immunization specifically protects older populations against SARS-CoV-2 variants but also highlight the importance of incorporating age as a key parameter in current and future vaccine design efforts[23].

## Methods

**Animals**. Female, 3-month-old BALB/c mice were purchased from the Jackson Laboratory. Female, 11-month-old BALB/c mice were purchased from Taconic Biosciences and used for aged mice experiments. Mice were housed under specific pathogen-free conditions at Boston Children's Hospital. All the procedures were approved under the Institutional Animal Care and Use Committee (IACUC) and operated under the supervision of the Department of Animal Resources at Children's Hospital (Protocol number 19-02-3897 R). At the University of Maryland School of Medicine, mice were housed in a biosafety level 3 facility for SARS-CoV-2 infections with all the procedures approved under the IACUC (Protocol #1120004).

**Immunization**. Mice were injected with BNT162b2 SARS-CoV-2 spike mRNA vaccine (Pfizer-BioNTech). BNT162b2 suspension (100 μg of mRNA/mL) was diluted 1:5 in phosphate-buffered saline (PBS) and 50 μL (1 μg) was injected intramuscularly in the caudal thigh. BNT162b2 was obtained as residual overfill volumes in used vials from the Boston Children's Hospital vaccine clinic, using only material that would otherwise be discarded, and was used within 6 h from the time of reconstitution. Mock treatment mice received PBS alone.

**SARS-CoV-2 wildtype spike and RBD expression and purification**. Full length SARS-CoV-2 Wuhan-Hu-1 spike glycoprotein (M1-Q1208, GenBank MN90894) and RBD constructs (amino acid residues R319-K529, GenBank MN975262.1), both with an HRV3C protease cleavage site, a TwinStrepTag and an 8XHisTag at C-terminus were obtained from Barney S. Graham (NIH Vaccine Research Center) and Aaron G. Schmidt (Ragon Institute), respectively. These mammalian expression vectors were used to transfect Expi293F suspension cells (Thermo Fisher)

using polyethylenimine (Polysciences). Cells were allowed to grow in 37 °C, 8% $CO_2$ for an additional 5 days before harvesting for purification. Protein was purified in a PBS buffer (pH 7.4) from filtered supernatants by using either StrepTactin resin (IBA) or Cobalt-TALON resin (Takara). Affinity tags were cleaved off from eluted protein samples by HRV 3C protease, and tag removed proteins were further purified by size-exclusion chromatography using a Superose 6 10/300 column (Cytiva) for full length Spike and a Superdex 75 10/300 Increase column (Cytiva) for RBD domain in a PBS buffer (pH 7.4).

**ELISA**. Spike protein-specific antibody concentrations were quantified in serum samples by ELISA by modification of a previously described protocol[28]. Briefly, high-binding flat-bottom 96-well plates (Corning) were coated with spike protein (25 ng per well) and incubated overnight at 4 °C. Plates were washed with 0.05% Tween 20/PBS and blocked with 1% bovine serum albumin (BSA)/PBS for 1 h at room temperature. Serum samples were serially diluted fourfold from 1:100 up to 1:1.05 × 10^8 and then incubated for 2 h at room temperature. Plates were washed three times and incubated for 1 h at room temperature with horseradish peroxidase (HRP)–conjugated anti-mouse IgG, IgG1, and IgG2a (SouthernBiotech). Plates were washed five times and developed with tetramethylbenzidine (BD OptEIA Substrate Solution, BD Biosciences) for 5 min and then stopped with 2 N $H_2SO_4$. Optical densities (ODs) were read at 450 nm with a SpectraMax iD3 microplate reader (Molecular Devices). Endpoint titers were calculated as the dilution that emitted an OD exceeding a 3x background. An arbitrary value of 50 was assigned to the samples with OD values below the limit of detection for which it was not possible to interpolate the titer.

**hACE2-RBD inhibition assay**. The hACE2-RBD inhibition assay used a modification of a previously published protocol[29]. Briefly, high-binding flat-bottom 96- well plates (Corning) were coated with recombinant hACE2 (100 ng per well) (Sigma-Aldrich) in PBS, incubated overnight at 4 °C, washed three times with 0.05% Tween 20 PBS, and blocked with 1% BSA PBS for 1 h at RT. Serum samples were diluted 1:160, pre-incubated with 3 ng of wildtype RBD-Fc in 1% BSA PBS for 1 h at RT, and then transferred to the hACE2-coated plate. RBD-Fc without preincubation with serum samples was added as a positive control, and 1% BSA PBS without serum preincubation was added as a negative control. Plates were then washed three times and incubated with HRP-conjugated anti-human IgG Fc (SouthernBiotech) for 1 h at RT. Plates were washed five times and developed with tetramethylbenzidine (BD OptEIA Substrate Solution, BD Biosciences) for 5 min and then stopped with 2 N $H_2SO_4$. The OD was read at 450 nm with a SpectraMax iD3 microplate reader (Molecular Devices). Percentage inhibition of RBD binding to hACE2 was calculated as: Inhibition (%) = [1 − (Sample OD value − Negative Control OD value)/(Positive Control OD value − Negative Control OD value)] × 100.

**Pseudovirus neutralization assay**. The SARS-CoV-2 pseudoviruses expressing a luciferase reporter gene were generated in an approach similar to as described previously[30,31]. Briefly, the packaging plasmid psPAX2 (AIDS Resource and Reagent Program), luciferase reporter plasmid pLenti-CMV Puro-Luc (Addgene), and spike protein expressing pcDNA3.1-SARS CoV-2 SΔCT of variants were co-transfected into HEK293T cells by lipofectamine 2000 (ThermoFisher). Pseudo-viruses of SARS-CoV-2 variants were generated by using WA1/2020 strain (Wuhan/WIV04/2019, GISAID accession ID: EPI_ISL_402124), B.1.617.2 (Delta, GISAID accession ID: EPI_ISL_2020950), or B.1.1.529 (Omicron, GISAID ID: EPI_ISL_7358094.2). The supernatants containing the pseudotype viruses were collected 48 h post-transfection, which were purified by centrifugation and filtration with 0.45 μm filter. To determine the neutralization activity of the serum samples, HEK293T-hACE2 cells were seeded in 96-well tissue culture plates at a density of $2 \times 10^4$ cells/well overnight. Three-fold serial dilutions of heat inactivated serum samples were prepared and mixed with 50 μl of pseudovirus. The mixture was incubated at 37 °C for 1 h before addition to HEK293T-hACE2 cells. After 48 h, cells were lysed in Steady-Glo Luciferase Assay (Promega) according to the manufacturer's instructions. SARS-CoV-2 neutralization titers were defined as the sample dilution at which a 50% reduction in relative light unit (RLU) was observed relative to the average of the virus control wells.

**Splenocyte restimulation, intracellular cytokine staining and flow cytometry**. Mouse spleens were mechanically dissociated and filtered through a 70 μm cell strainer. After centrifugation, cells were treated with 1 mL ammonium-chloride-potassium lysis buffer for 2 min at RT. Cells were washed and plated in a 96-well U-bottom plate ($2 \times 10^6$/well) and incubated overnight in RPMI 1640 supplemented with 10% heat-inactivated FBS, penicillin (100 U/ml), streptomycin (100 mg/ml), 2-mercaptoethanol (55 mM), non-essential amino acids (60 mM), HEPES (11 mM), and L-Glutamine (800 mM) (all Gibco). Next day, SARS-CoV-2 wildtype (PM-WCPV-S-1) or Omicron (PM-SARS2-SMUT08-1) spike peptide pools (JPT) were added at 0.6 nmol/ml in the presence of anti-mouse CD28/49d (1 μg/mL, BD) and brefeldin A (5 μg/ml, BioLegend). After 6 h stimulation, cells were washed twice and were treated with Mouse Fc Block (BD) according to the manufacturer's instructions. Cells were washed and stained with Aqua Live/Dead stain (Life Technologies, 1:500) for 15 min at RT. Following two additional washes, cells were incubated with the following Abs for 30 min at 4 °C: anti-mouse CD44

[IM7, PerCP-Cy5.5, BioLegend #103032, 1:160], anti-mouse CD3 [17A2, Brilliant Violet 785, BioLegend #100232, 1:40], anti-mouse CD4 [RM4-5, APC/Fire 750, BioLegend 100568, 1:160] and anti-mouse CD8 [53-6.7, Brilliant UltraViolet 395, BD #563786, 1:80]. Cells were then fixed and permeabilized by using the BD Cytofix/Cytoperm kit according to the manufacturer's instructions, and were subjected to intracellular staining (30 min at 4 °C) using the following Abs: anti-mouse IFNγ [XMG1.2, Alexa Fluor 488, BioLegend #505813, 1:160], anti-mouse TNF [MP6-XT22, PE Cy7, BioLegend # 506324, 1:160], anti-mouse IL-2 [JES6-5H4, PE, BioLegend # 503808, 1:40]. Finally, cells were fixed in 1% paraformaldehyde (Electron Microscopy Sciences) for 20 min at 4 °C and stored in PBS at 4 °C until acquisition. Samples were analyzed on an LSR II (BD) flow cytometer and FlowJo v10.8.1 (FlowJo LLC).

**SARS-CoV-2 Omicron challenge study**. Mice were anesthetized interperitoneally with 50 μL ketamine (1.3 mg/mouse) and xylazine (0.38 mg/mouse) diluted in PBS. Mice were then intranasally inoculated with $1 \times 10^5$ PFU of SARS-CoV-2 Omicron (BA.1), courtesy of Dr. Mehul Suthar, in 50 μL of PBS divided between nares. Challenged mice were weighed daily. On day 2 post-infection, the mice were euthanized, and the lungs were harvested for histology, analysis of host responses by qPCR, and determination of viral titer by plaque assay.

**SARS-CoV-2 plaque assay**. The day prior to infection, 2.5e5 VeroTMPRSS2 cells were seeded per well in a 12-well plate in 1 mL of VeroTMPRSS2 media [DMEM (Quality Biological), 10% FBS (Gibco), 1% Penicillin-Streptomycin (Gemini Bio Products), and 1% L-Glutamine (Gibco)]. Tissue samples were thawed and homogenized with 1 mm beads in a Bead ruptor (Omni International Inc.) and then spun down at 21,000 g for 2 min. A 6-point dilution curve was prepared by serial diluting 25 μL of sample 1:10 in 225 μL DMEM. 200 μL of each dilution was then added to the cells and the plates were rocked every 15 min for 1 h at 37 °C. After 1 h, 2 mL of a semi-solid agarose overlay was added to each well [DMEM, 4% FBS, and 0.06% UltraPure agarose (Invitrogen)]. After 72 h at 37 °C and 5% $CO_2$, plates were fixed in 2% PFA for 20 min, stained with 0.5 mL of 0.05% Crystal Violet and 20% EtOH, and washed twice with $H_2O$ prior to counting of plaques. The titer was then calculated. For tissue homogenates, this titer was multiplied by 40 based on the average tissue sample weight being 25 mg.

**Gene expression analysis by qPCR**. RNA was isolated from TRI Reagent samples using phenol-chloroform extraction or column-based extraction systems (Direct-zol RNA Miniprep, Zymo Research) according to the manufacturer's protocol. RNA concentration and purity (260/280 and 260/230 ratios) were measured by NanoDrop (Thermo Fisher Scientific). Samples with an A260/A280 ratio of <1.7 were excluded for further analysis. cDNA was prepared from purified RNA with $RT^2$ First Strand Kit, per the manufacturer's instructions (Qiagen). cDNA was quantified by qPCR on a 7300 real-time PCR system (Applied Biosystems) using pre-designed SYBR Green Primers (QIAGEN) specific for Ifit2 (PPM05993A), Rsad2 (PPM26539A), and Rpl13a (PPM03694A).

**Statistics and reproducibility**. Mouse experiments aimed to include in total 20 mice per group and were from single experiments. Sample size and age criteria were chosen empirically based on the results of previous studies. Mice were randomly assigned to different treatment groups. No data outliers were excluded. Statistical analyses were performed using Prism v9.0.2 (GraphPad Software). Two-tailed P values <0.05 were considered significant. Normally distributed data were analyzed by one-way analyses of variance (ANOVAs) corrected for multiple comparisons. Non-normally distributed data were log-transformed and analyzed by ANOVA or analyzed by Kruskal–Wallis test corrected for multiple comparisons.

**Reporting summary**. Further information on research design is available in the Nature Research Reporting Summary linked to this article.

## Data availability

The authors declare that all data supporting the findings of this study are available within the supplementary information files (Supplementary Data 1).

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

## Acknowledgements

The current study was supported, in part, by U.S. National Institutes of Health (NIH)/National Institutes of Allergy and Infectious Diseases (NIAID) awards, including Adjuvant Discovery (HHSN272201400052C and 75N93019C00044) and Development (HHSN272201800047C) Program Contracts, a Massachusetts Consortium on Pathogen Readiness (Mass-CPR) award, as well as philanthropic support from Amy and Michael Barry to O.L.; NIH grant (1R21AI137932-01A1) and Adjuvant Discovery Program contract (75N93019C00044) to D.J.D.; BARDA #ASPR-20-01495, DARPA #ASPR-20-01495, NIH R01 AI148166, and NIH HHSN272201400007C to M.B.F. The Precision Vaccines Program is supported, in part, by the BCH Department of Pediatrics and the Chief Scientific Office. We thank the members of the BCH *Precision Vaccines Program* (PVP) for helpful discussions as well as K. Churchwell, G. Fleisher, D. Williams, and A. Cervini for support of the PVP. Work within the PVP on this project was funded, in part, by philanthropic support from Amy and Michael Barry, Stop & Shop, and the Boston Investment Conference via the BCH Trust. We thank B. S. Graham (NIH Vaccine Research Center) for providing the plasmid for prefusion stabilized SARS-CoV-2 spike trimer. We thank the pharmacists of Boston Children's Hospital for efforts to maximize the use of SARS-CoV-2 vaccines by saving leftover or overfill of otherwise-to-be-discarded vaccine vials. E.N. is a JSPS Overseas Research Fellow and a joint Society for Pediatric Research and Japanese Pediatric Society Scholar. D.J.D. would like to thank S. McHugh, G. Boyer, L. Conetta and the staff of Lucy's Daycare, the staff YMCA of Greater Boston, Bridging Independent Living Together (BILT) Inc., and the Boston Public Schools for childcare and educational support during the COVID-19 pandemic. The graphics in Fig. 1 and Supplementary Fig. 1a were created with BioRender.

## Author contributions

E.N. and M.E.M. conceived, designed, performed, and analyzed the experiments and wrote the paper. T.R.O. performed mouse serological assays and splenocyte restimulation and their analysis. M.M. performed splenocyte restimulation. S.B. performed splenocyte restimulation, flow cytometry assay, and analyzed the data. C.D. and M.B.F. performed and analyzed SARS-CoV-2 Omicron mouse challenge study. J.Y., H.W., and D.H.B. performed and analyzed pseudovirus neutralization experiments. K.S., A.Z.X., L.S., H.-S.S., and S.D.-P. expressed and purified SARS-CoV-2 spike protein. B.B., F.B., A.-N.B., and R.K.E. contributed to the design of experiments. O.L., M.B.F., and D.J.D. conceived the project, designed the experiments, and wrote the manuscript.

## Competing interests

The authors declare the following competing interests: O.L. has served as a paid consultant to Moody's Analytics and the Midsized Bankers Association of America; M.B.F. serves on the scientific advisory board of Aikido Pharma and has collaborative research agreements with Novavax, AstraZeneca, Regeneron and Irazu Bio; E.N., T.R.O., F.B., O.L., and D.J.D. are named inventors on vaccine adjuvant patents. These commercial or financial relationships are unrelated to the current study. The remaining authors declare no competing interests.
