## [Peer Review File · Communications Biology]

Reviewers' comments:

Reviewer #1 (Remarks to the Author):

In this study, the authors investigate vaccine strategies to enhance immunogenicity and protection against SARS-CoV-2 in aged mice. The manuscript is clearly written. The authors conclude that boosting of aged animals induces robust cellular and humoral immune responses that lead to protection against rechallenge. The major points of the manuscript are scientifically sound and largely supported by the data, albeit with the acknowledged caveats of limited sample size in the aged cohort. This is one major concern limiting enthusiasm for this manuscript. The other major concern is the source of aged mice and young adult mice are from different vendors, further complicating interpretations especially in comparisons across age.

The use of 'extreme' in relation to mouse age is purely subjective and imparts no meaningful information to the reader. Moreover, it is not a commonly used descriptive for stages of aging in the literature. Generally, mice older than 18 months are considered old or aged. Mice 11 months old should be considered aging or middle-aged. Terminology should be updated throughout the manuscript to match best practices for the field.

Figure 1 is very confusing given that there is a sole figure meant to outline the study design. The relationship from treatment group and age group is unclear. Does each treatment group have a corresponding age group associated with it? This is clearly the case in subsequent figures but is unclear as presented in Figure 1. Perhaps it would be clearer if a separate schema for each experimental and control group is shown. Additionally, please use consistent titles and designations throughout. Figure 1 uses "booster vax", figure 2A-D uses "Boost", figure 2E uses "Booster" etc.

In Figure S2, CD44 gating on CD8 T cells is overly strict and ~15% of the responding population is being excluded from analysis. While this should not alter any interpretations, the cytokine production among CD44^{int/hi} should be included in the analysis throughout.

Some numbers are not clear in the figures. For example, Figure 2E GMTs are not clear. It appears the fold changes are on top of the GMTs.

Figure 4 title is slightly misleading. "mRNA booster vaccination is essential to protect" is not shown as distal vaccination also offers some protection. Similar language is used in the text. While it is clear that boosting offers superior protection it is not essential as distal vaccination offers some protection from disease.

A major case to claim that boosting is superior to distant vax in 12 mo is the amount of "sterilizing" immunity shown indirectly in Figure 4A. This should be graphically represented in some fashion to better emphasize this point. For example, a simple chart reflecting the % of samples above the limit of detection would impart important information to the reader.

Were correlations of protection assessed for CD4 and CD8 T cells?

The authors make bold claims in the discussion that need important caveats. For example, the first line of the discussion states "In this study[,], we have demonstrated for the first time the age-specific impact of booster immunization on immunogenicity and protection in a murine model." Authors should modify the end of the sentence to "in a murine model of SARS-CoV-2 infection. Concerning Covid, others have reported boosting strategies enhance immunogenicity in aged mice (Silva-Cayetano et al., Med (NY); 2021). This study should be discussed as it uses an adenoviral covid vaccine suggesting that the impact of boosting on aged populations is not platform specific.

Are there any additional readouts of disease? For example, can weight loss be assessed in this short time period after challenge? Or is lung pathology visible? Essentially, while the viral burden is reduced, does this come at the cost of enhanced immunopathology? This should at least be discussed as viral titers are just one readout.

Reviewer #2 (Remarks to the Author):

Age impacts the response to many vaccinations including a SARS-CoV-2 mRNA vaccination. This manuscript presents a straightforward study examining the ability of a SARS-CoV-2 booster vaccination to enhance the response in older previously vaccinated mice. The authors do a nice job of examining the humoral and T cell responses induced by the vaccine and then go on to assess protection from challenge. It is very clear from their data that a booster vaccination can enhance both humoral and T cell responses in older mice.

This manuscript is well written and the results are presented in a clear and concise manner. There are only a couple of minor points that the authors should address:

1. 11 month old mice are not "aged", they are middle aged and are equivalent to a human of approximately 40 years of age (according to the Jackson laboratory website). Since mice of multiple ages are used in this study, the best way to present this would be to refer to the actual ages of the mice (which the authors do frequently) instead of "young", "aged" or "extremely aged".
2. There are no axis labels on flow plots in supplementary figure 2.

Reviewer #2:

Age impacts the response to many vaccinations including a SARS-CoV-2 mRNA vaccination. This manuscript presents a straightforward study examining the ability of a SARS-CoV-2 booster vaccination to enhance the response in older previously vaccinated mice. The authors do a nice job of examining the humoral and T cell responses induced by the vaccine and then go on to assess protection from challenge. It is very clear from their data that a booster vaccination can enhance both humoral and T cell responses in older mice.

This manuscript is well written and the results are presented in a clear and concise manner. There are only a couple of minor points that the authors should address:

We appreciate the Reviewer's positive assessment of our work and constructive suggestions. We have now further revised and improved the manuscript accordingly as outlined below.

Minor concerns

1. 11 month old mice are not “aged”, they are middle aged and are equivalent to a human of approximately 40 years of age (according to the Jackson laboratory website). Since mice of multiple ages are used in this study, the best way to present this would be to refer to the actual ages of the mice (which the authors do frequently) instead of “young”, “aged” or “extremely aged”.

We have amended these words accordingly in the updated manuscript. As per the Reviewer's suggestion, the manuscript now cites the age in question for each experimental design and readout.

2. There are no axis labels on flow plots in supplementary figure 2.

Thank you for the careful review. To address this point, we added axis labels in the resubmitted Supplementary Figure 2.

Reviewer #1:

In this study, the authors investigate vaccine strategies to enhance immunogenicity and protection against SARS-CoV-2 in aged mice. The manuscript is clearly written. The authors conclude that boosting of aged animals induces robust cellular and humoral immune responses that lead to protection against rechallenge. The major points of the manuscript are scientifically sound and largely supported by the data, albeit with the acknowledged caveats of limited sample size in the aged cohort. This is one major concern limiting enthusiasm for this manuscript.

We thank the Reviewer for this comment and acknowledge the limitation of the small sample size in 21-month-old mice. As stated in the initial manuscript, some of the experimental groups had a relatively low number of mice, mainly due to the long term/longitudinal nature of the study design. Regarding the aged mice cohort, we initially included N=80 of 11-month-old mice at study week 0 (N=20 per treatment group). However, over the 10 months period of this study, several mice developed serious conditions such as frailty, severe dermatitis, vaginal prolapse, and gross tumor. We thus had to euthanize a considerable number of mice adhering to animal ethics in communication with the staff of animal care resources at Boston Children's Hospital. As a result, the initial N=80 mice decreased to N=58 at week 36 for Ab measurement, and only N=48 of 21-month-old mice could be assessed for T cell responses (N=20) or challenge study (N=28). Given the importance and timeliness of this manuscript, we could not repeat this experiment because it would take up to one year (possibly more) to increase the number of animals.

Nonetheless, our study demonstrates the importance of booster vaccination in the aged, 21-month-old mice for the following reasons. First, the sample size for Ab analysis, the most established immune correlate of protection for SARS-CoV-2, is ample (N=13–17 per group). Second, although the number of 21-month-old mice was limited in T cell study (N=4–6 per group), the effect size of the booster dose was significant (4.3- and 12.9-fold increase of specific CD4⁺ and CD8⁺ T cell responses, respectively; $P < 0.001$ and < 0.0001), and there was no data overlap between “21M Distant vax” and “21M Booster” groups. Lastly, the challenge study also demonstrated significant effect size of booster dose in the limited number of 21-month-old mice (N=5-9 per group). As in the Figure 4A, 67% (4/6) of “21M Distant vax” mice showed detectable viral loads while 0% (0/5) of the “21M Booster” mice groups showed detectable live virus in the lung ($P=0.06$, Fisher's exact test).

To address the Reviewer's concern, we have added underlined text to the revised manuscript (lines 238–243):

“Second, due to the unique and extremely longitudinal nature of the study design, some of the experimental groups had a relatively low number of mice, restricting some statistical comparisons. Of note, a considerable number of aged mice developed conditions such as frailty and gross tumors over the 10-month study period, and only N=48 out of the initial N=80 could be assessed for T cell responses (N=20) or challenge study (N=28). Nonetheless, due to the ample sample size for Ab analysis (N=13–17 per group) and large effect size of the booster dose observed in T cell and challenge studies, the results of this study demonstrate the importance of booster vaccination in the aged 21-month-old mice.”

The other major concern is the source of aged mice and young adult mice are from different vendors, further complicating interpretations especially in comparisons across age.

As pointed out by the Reviewer, there is a vendor difference in addition to age difference between initially 3- and 11-month-old mice. Due to the limited number of mice in the vendors at the start of this experiment, we purchased 3-month-old BALB/c mice from the Jackson Laboratory (Jax) and 11-month-old BALB/c mice from Taconic. During this longitudinal study, n=40 of initially 11-month-old mice purchased from Taconic received primary vaccination series at week 0, and n=20 of initially 3-month-old mice purchased from Jax received primary vaccination series at week 32 at the age of 11-month-old. We therefore compared immunogenicity between Jax and Taconic mice which both received primary vaccination series at 11-month-old age.

As shown in the new Supplementary Figure 3, anti-spike IgG titers measured 2 weeks after primary vaccination series demonstrated that the immunogenicity between 11-month-old Jax and Taconic mice were similar while considerable difference was observed vs 3-month-old Jax mice. Further, 11-month-old Taconic mice demonstrated slightly lower anti-spike IgG titer as compared to 11-month-old Jax mice (GMT: 11915 vs 19525). As such, the difference between 3M Jax and 12M Taconic is likely smaller than the difference between 3M Jax and 12M Jax mice; indicating that the age-specific differences observed were not due to different vendors.

We added the following sentences to the revised manuscript (lines 243–252) and new Supplementary Figure 3:

“Third, due to limited availability, 3- and 11-month-old mice were purchased from different vendors. During this longitudinal study, N=40 of initially 11-month-old mice received a primary vaccination series at week 0, while N=20 of initially 3-month-old mice received a primary vaccination series at week 32 at the age of 11-month-old. We therefore compared immunogenicity between mice purchased from two vendors which received primary vaccination series at 11-month-old age and confirmed that the immunogenicity of 11-month-old mice from the two vendors were largely equivalent and both considerably different compared to 3-month-old mice (Supplementary Fig. 3). Accordingly, any vendor-related differences do not account for any of the key interpretations in our manuscript.”

Supplementary Figure 3. 11-month-old BALBc mice demonstrate similar anti-spike IgG responses after primary mRNA vaccination series

3-month-old BALB/c mice acquired from the Jackson Laboratory (Jax) received primary mRNA vaccination series at either study week 0 at the age of 3-month ("3M_Jax") or study week 32 at the age of 11-month ("11M_Jax"). 11-month-old BALB/c mice acquired from Taconic and received primary vaccination series at study week 0 ("11M_Taconic"). Sera were collected 2 weeks after primary vaccination series and anti-spike IgG titers were determined. Each symbol represents an individual sample and red horizontal lines and numbers represents geometric mean titers (GMTs). Dashed lines represent lower limit of detection. N=40, 39 and 19 per group for "3M_Jax", "11M_Jax", and "11M_Taconic", respectively. Data were analyzed by Kruskal-Wallis test corrected for multiple comparisons. **** $P < 0.0001$.

Minor concerns

1. The use of 'extreme' in relation to mouse age is purely subjective and imparts no meaningful information to the reader. Moreover, it is not a commonly used descriptive for stages of aging in the literature. Generally, mice older than 18 months are considered old or aged. Mice 11 months old should be considered aging or middle-aged. Terminology should be updated throughout the manuscript to match best practices for the field.

As suggested, we have amended these words accordingly in the updated manuscript.

2. Figure 1 is very confusing given that there is a sole figure meant to outline the study design. The relationship from treatment group and age group is unclear. Does each treatment group have a corresponding age group associated with it? This is clearly the case in subsequent figures but is unclear as presented in Figure 1. Perhaps it would be clearer if a separate schema for each experimental and control group is shown. Additionally, please use consistent titles and designations throughout. Figure 1 uses "booster vax", figure 2A-D uses "Boost", figure 2E uses "Booster" etc.

We thank the Reviewer for this comment. Accordingly, we have updated Figure 1 to clarify the relationship between mouse age groups and treatments. Furthermore, we have amended terms accordingly in the updated manuscript.

Figure 1. Study design outlining the longitudinal immunization of mRNA BNT162b2 across the lifespan

3. In Figure S2, CD44 gating on CD8 T cells is overly strict and ~15% of the responding population is being excluded from analysis. While this should not alter any interpretations, the cytokine production among CD44^{int/hi} should be included in the analysis throughout.

We thank the Reviewer for the careful review. We have amended CD44 gating on both CD4⁺ and CD8⁺ T cells (see new Supplementary Figure 2) and updated data in Figure 3 and related text.

Supplementary Figure 2. Ex vivo flow cytometry gating strategy post murine mRNA vaccination

4. Some numbers are not clear in the figures. For example, Figure 2E GMTs are not clear. It appears the fold changes are on top of the GMTs.

To address this point, we have removed fold change values from Figure 2E.

5. Figure 4 title is slightly misleading. “mRNA booster vaccination is essential to protect” is not shown as distal vaccination also offers some protection. Similar language is used in the text. While it is clear that boosting offers superior protection it is not essential as distal vaccination offers some protection from disease.

We thank the Reviewer for this comment. We have amended Figure 4 title and related text accordingly.

6. A major case to claim that boosting is superior to distant vax in 12 mo is the amount of “sterilizing” immunity shown indirectly in Figure 4A. This should be graphically represented in some fashion to better emphasize this point. For example, a simple chart reflecting the % of samples above the limit of detection would impart important information to the reader.

We thank the Reviewer for this comment and agree regarding the value of a figure to emphasize the protection induced by a boost in 21-month-old mice. To this end, we added pie charts demonstrating the proportion of mice within each group that had detectable virus in the lungs (see new Figure 4A).

7. Were correlations of protection assessed for CD4 and CD8 T cells?

We analyzed correlation between lung viral titers and immunogenicity data collected from each individual mouse. In other words, we were limited to analyzing correlates of protection from mice that underwent the challenge study. To measure T cell responses in mice, we had to euthanize mice and collect splenocytes at week 36 (prior to the challenge study). Thus, we only assessed Ab data for the correlation study.

To make this limitation clearer, we have added underlined words to the revised manuscript (lines 223–224):

“Since we had to euthanize mice and collect splenocytes to assess T cell response, our correlation analysis was limited to readouts of humoral immunity. However, T cells may play an important role in protecting against Omicron. Indeed, a few mice demonstrated protection from lung infection without detectable neutralizing activity. These findings suggest that protection against Omicron may correlate with both humoral and cellular immune responses.”

8. The authors make bold claims in the discussion that need important caveats. For example, the first line of the discussion states “In this study[,] we have demonstrated for the first time the age-specific impact of booster immunization on immunogenicity and protection in a murine model.” Authors should modify the end of the sentence to “in a murine model of SARS-CoV-2 infection. Concerning Covid, others have reported boosting strategies enhance immunogenicity in aged mice (Silva-Cayetano et al., Med (NY); 2021). This study should be discussed as it uses an adenoviral covid vaccine suggesting that the impact of boosting on aged populations is not platform specific.

To address the Reviewer’s point, we amended the sentence accordingly. Further, we have added a sentence to the revised manuscript (lines 194–196):

“Additionally, boosting strategies demonstrated enhanced immunogenicity in aging and aged mice in other SARS-CoV-2 vaccine platforms, including adenoviral vector and adjuvanted protein subunit vaccines^{22,23}.”

9. Are there any additional readouts of disease? For example, can weight loss be assessed in this short time period after challenge? Or is lung pathology visible? Essentially, while the viral burden is reduced, does this come at the cost of enhanced immunopathology? This should at least be discussed as viral titers are just one readout.

We thank the Reviewer for this important suggestion. We euthanized mice at day 2 post infection, as this day was the best time point to analyze lung viral titers based on our preliminary experiments. However, it is difficult to analyze body weight loss and lung pathology this close to the initial infection. Nevertheless, we agree with the Reviewer regarding the importance of analyzing host immune responses. We therefore investigated expression of IFN stimulated genes (ISGs) in the lung and demonstrated that a booster dose contributes to protect 21-month-old mice from infection-induced immune responses.

We have added the new Figure 4B and following sentences to the revised manuscript (lines 159–163):

“Type I interferons are significant drivers of pathological responses after SARS-CoV-2 infection¹². We therefore assessed the expression of IFN-stimulated genes (ISGs) and confirmed a decrease in the host antiviral response in the 21-month-old “booster” mice

compared to the 21-month-old "distant vax" mice which would correlate to the lower viral load in the lungs (Fig 4A-B)."

REVIEWERS' COMMENTS:

Reviewer #1 (Remarks to the Author):

The authors have addressed all my concerns. I appreciate the authors dilemma with aging and mice. I believe it would be inappropriate to hold up publication in order to increase the number of mice at the oldest age group given the acknowledgement and discussion by the authors.

Newly included data alleviates some concerns with vendor source. However, authors should temper their conclusions in newly added discussion. "...vendor-related differences **do not** account for any of the key interpretations..." should be changed to "vendor-related differences **are unlikely** to account for any of the key interpretations..." since readouts other than antibody are reported throughout the paper and the new supplemental data only reports that antibody is similar between vendors.

Reviewer #1:

The authors have addressed all my concerns. I appreciate the authors dilemma with aging and mice. I believe it would be inappropriate to hold up publication in order to increase the number of mice at the oldest age group given the acknowledgement and discussion by the authors.

We appreciate the Reviewer's positive assessment of our work and constructive suggestions. We have now further revised and improved the manuscript accordingly as outlined below.

Newly included data alleviates some concerns with vendor source. However, authors should temper their conclusions in newly added discussion. "...vendor-related differences do not account for any of the key interpretations..." should be changed to "vendor-related differences are unlikely to account for any of the key interpretations..." since readouts other than antibody are reported throughout the paper and the new supplemental data only reports that antibody is similar between vendors.

We have amended these words accordingly in the updated manuscript.